# Independent-Set Design of Experiments for Estimating Treatment and Spillover Effects under Network Interference

**Chencheng Cai**[1][*][†]    **Xu Zhang**[2][*]    **Edoardo M. Airoldi**[3]
[1]Department of Mathematics and Statistics, Washington State University    [2]Amazon
[3]Department of Statistics, Operations, and Data Science, Temple University
chencheng.cai@wsu.edu   xuzhangn@amazon.com   airoldi@temple.edu

## Abstract

Interference is ubiquitous when conducting causal experiments over networks. Except for certain network structures, causal inference on the network in the presence of interference is difficult due to the entanglement between the treatment assignments and the interference levels. In this article, we conduct causal inference under interference on an observed, sparse, but connected network, and we propose a novel design of experiments based on an independent set. Compared to conventional designs, the independent-set design focuses on an independent subset of data and controls their interference exposures through the assignments to the rest (auxiliary set). We provide a lower bound on the size of the independent set from a greedy algorithm and justify the theoretical performance of estimators under the proposed design. Our approach is capable of estimating both spillover effects and treatment effects. We justify its superiority over conventional methods and illustrate the empirical performance through simulations.

## 1 Introduction

Randomized experiments are widely regarded as the gold standard for estimating causal effects. However, the spillover effect is ubiquitous in experiments when one unit's treatment affects others' outcomes, where the stable unit treatment value assumption (SUTVA) (Imbens & Rubin, 2015) is violated. Examples include the traditional agricultural field experiments (Zaller & Köpke, 2004), educational studies (Rosenbaum, 2007), econometric assumptions (Banerjee et al., 2013; Johari et al., 2020), and social networks (Phan & Airoldi, 2015). The spillover effect is also called "interference", and the two names are used interchangeably in the literature.

Estimating causal effects in the presence of interference is usually difficult, except for certain special cases. For instance, when the population of units can be well-partitioned into several isolated clusters, randomized saturation design (Hudgens & Halloran, 2008) and its variations (for example, Eckles et al. (2017), Owen & Varian (2020)) has shown success in estimating causal effects involving interference by controlling the interference level at each cluster with a pre-determined saturation. The difficulties in estimating spillover effects on a well-connected network are two-fold. The first is the entanglement between the treatment assignments and the spillover effects received by the units. Because the spillover effects received by the units are determined by the interference relationships and the assignment of all units, one cannot achieve separate controls on the treatment and the spillover of all units. The second difficulty comes from the collapse of the spillover effect. A completely randomized experiment on all units anchors the spillover effect received by each unit at its expectation for large networks because of the law of large numbers. In estimating spillover effects or total treatment effects, where we prefer diversified interference levels for the units, the collapse gives rise to an increase in variance and a reduction in power.

Although the individual treatment assignments and the interference received cannot be controlled independently for the whole dataset, an approximately separate control is feasible for a subset of

---

[*]Equal contribution.
[†]Corresponding author.

the dataset. We propose a novel design of experiments on the network, where we focus on a high-quality subset of units. The subset contains non-interfering units and is called the "independent set", where we borrow the name from graph theory to emphasize that there is no edge in the independent set. The rest of the units form the "auxiliary set", which provides interference to the units in the independent set. The partitioning of the independent and auxiliary sets re-gains separate controls over the treatment and the interference on the independent set. The approach can be viewed as a trade-off between the quantity and the quality of data. The independent set has a smaller sample size compared to the full dataset. However, due to the possibility of separate controls of their treatments and interference, observed outcomes from the independent set units provide more information for the causal effect and, therefore, are more efficient for causal inference purposes.

This paper is organized as follows: Section 2 discusses related studies and summarizes our contribution. We formally define the causal effect under interference and introduce the independent set design in Section 3. Section 4 provides the theoretical foundation and quantifies the performance. In Section 5, we evaluate the performance of our approach on simulated data. In Section 6, we discuss and highlight potential future work. Theorem proofs are provided in the Appendix.

## 2 RELATED WORK AND OUR CONTRIBUTION

In classical methods for randomized experiments, a key assumption is that there is no interference among the units, and the outcome of each unit does not depend on the treatments assigned to the other units (SUTVA). However, this no-interference assumption is not plausible on networks. Many approaches are proposed to address the interference problem. First, researchers improve experimental designs to reduce interference, such as the cluster-based randomized design (Bland, 2004), group-level experimental designs (Sinclair et al., 2012; Basse et al., 2019), randomized saturation design (Hudgens & Halloran, 2008), and graph cluster randomization (Ugander et al., 2013; Eckles et al., 2017; Ugander & Yin, 2023). These approaches aim to reduce interference by clustering and assume that interference only exists within units in the cluster and that there is no interference between clusters (known as partial interference (Sobel, 2006)). However, when the isolated cluster assumption is violated, the randomized saturation design fails to estimate effects accurately (Cai et al., 2022). To extend the cluster-based randomized design, ego-clusters design (Saint-Jacques et al., 2019) is proposed to estimate spillover effects, which requires non-overlapping ego-clusters. Karwa & Airoldi (2018) introduced a simple independent set design, which only focuses on estimating treatment effect and has no control of bias/variance of the estimator. Unlike graph cluster randomization and ego-cluster randomization, which require non-overlapping clusters, our design allows units to share neighbors. Furthermore, recent studies incorporate network structure in the experiment based primarily on the measurement of local neighbors of units (Awan et al., 2020), and they estimate causal effects by matching the units with similar network structures. Some studies use special network structures in designs to relax SUTVA assumption and estimate causal effects (Aronow & Samii, 2017; Toulis & Kao, 2013; Yuan et al., 2021). In another direction, some studies leverage inference strategies, and they conduct a variety of hypotheses under network inference (Athey et al., 2018; Rosenbaum, 2007; Pouget-Abadie et al., 2019b; Sussman & Airoldi, 2017). When there exists a two-sided network or two-sided market for units, for example, consumers and suppliers, researchers proposed bipartite experiments (Pouget-Abadie et al., 2019a; Holtz et al., 2020; Doudchenko et al., 2020; Zigler & Papadogeorgou, 2021) to mitigate spillover effects. In bipartite experiments, two distinct groups of units are connected together and form a bipartite graph. In contrast, our approach generates two sets from units that belong to one group in a network. In addition, an increasing number of studies focus on observational studies on networks (Liu et al., 2016; Ogburn & VanderWeele, 2017; Forastiere et al., 2021). However, these methods require structural models and need to identify confounders.

Our method makes several key contributions: First, we propose a novel design that partitions data into two disjoint sets to gain separate controls on the treatment and the interference. Second, in contrast to most previous studies, which only focus on estimating the treatment effect or spillover effect, our design works as a general framework for network experiments involving interference and is capable of various causal tasks (estimating both treatment and spillover effects). Third, the treatment assignments on the two sets can be optimized directly to control the bias/variance of the estimator. The connection between the objective function and the bias/variance is discussed in Section 4. Fourth, we provide theoretical guarantees, including an almost-sure lower bound on

Table 1: Comparison of designs for estimating spillover effects

| Design | Network Assumption | Sample Size* | Interference Control |
|---|---|---|---|
| **Independent Set** | random, sparse | $(n \log s)/s$ | high |
| Ego-clusters | random, sparse | $n/(s+1)$ | full |
| Randomized Saturation | isolated clusters | $n$ | partial |
| Completely Randomized | no | $n$ | no |

∗: number of units used for inference, $n$: network size, $s$: average degree

the sample size and the analytic bias/variance for the estimators. Finally, unlike previous studies, which require SUTVA assumptions and no interference between units or are restricted to special network structures to mitigate the spillover effects, our approach does not require SUTVA and could be implemented in arbitrary networks.

In Table 1, we compare our method to a few competitive designs in terms of assumptions on networks, effective sample size, and the ability to control interference.

## 3 INDEPENDENT-SET DESIGN

### 3.1 POTENTIAL OUTCOMES AND CAUSAL EFFECTS UNDER INTERFERENCE

Consider a set of $n$ experimental units labeled with $i = 1, \ldots, n$. Each of them is assigned with a treatment $Z_i \in \{0, 1\}$, for which we denote $Z_i = 1$ for treated, $Z_i = 0$ for control, and $\boldsymbol{Z} := (Z_1, \ldots, Z_n)$ for the full treatment vector. Under Rubin's framework of potential outcomes (Imbens & Rubin, 2015), we assume $Y_i(\boldsymbol{Z})$ is the potential outcome of unit $i$ that would be observed if the units **were** assigned with the treatment vector $\boldsymbol{Z}$. We say "unit $i$ is interfered by unit $j$" when $Y_i$ depends on $Z_j$, for $i \neq j$. All the pairwise interference relationships can be represented by a graph $\mathcal{G} = (V, E)$, where $V = \{1, \ldots, n\}$ is the vertex set, and $(i, j) \in E$ if units $i$ and $j$ interfere with each other. Here, we assume all pairwise interference is symmetric such that $\mathcal{G}$ can be reduced to an undirected graph with each undirected edge representing the original bi-directed edges. The left panel in Figure 1 shows a diagram of 12 units and 18 interfering pairs.

Given the graph $\mathcal{G}$, we write the unit $i$'s potential outcome as $Y_i(Z_i, \boldsymbol{Z}_{\mathcal{N}_i})$, where $\mathcal{N}_i := \{j \in V : (i, j) \in E\}$ is the neighbor of unit $i$, and $\boldsymbol{Z}_{\mathcal{N}_i} := (Z_j)_{j \in \mathcal{N}_i}$ is the neighbor treatment (sub-)vector. Following Forastiere et al. (2021); Cai et al. (2022), we further assume that $Y_i$ depends on unit $i$'s neighbor through the proportion of treated neighbors, which is defined by

$$\rho_i := |\mathcal{N}_i|^{-1} \sum_{j \in \mathcal{N}_i} Z_j. \tag{1}$$

Assumption 1 assumes the interference on the proportion of treated neighbor assumption. In this paper, we simply write $Y_i(Z_i, \rho_i) : \{0, 1\} \times [0, 1] \to \mathbb{R}$ as the potential outcome of unit $i$.

**Assumption 1 (interference on the proportion of treated neighbors)** *Let $\boldsymbol{z}$ and $\boldsymbol{z}'$ be two treatment vectors, and let $Y_i(\boldsymbol{Z})$ be the potential outcome of unit $i$. We assume*

$$Y_i(\boldsymbol{z}) = Y_i(\boldsymbol{z}') \quad \text{whenever } z_i = z_i' \text{ and } \rho_i = \rho_i',$$

*where $\rho_i$ and $\rho_i'$ are the proportion of treated neighbors induced from $\boldsymbol{z}$ and $\boldsymbol{z}'$, correspondingly.*

Furthermore, for any two tuples $(z, \rho), (z', \rho') \in \{0, 1\} \times [0, 1]$, we define the unit-level causal effect as $\tau_i(z, \rho, z', \rho') := Y_i(z, \rho) - Y_i(z', \rho')$, with certain special cases defined as

| | | |
|---|---|---|
| (direct effect) | $\tau_i^{(d)}(\rho)$ | $:= \tau_i(1, \rho, 0, \rho),$ |
| (spillover effect) | $\tau_i^{(i)}(z, \rho, \rho')$ | $:= \tau_i(z, \rho, z, \rho'),$ |
| (total effect) | $\tau_i^{(t)}$ | $:= \tau_i(1, 1, 0, 0).$ |

The direct effect measures the marginal effect of $z_i$ under a given level of interference. The indirect effect relates to the effect of interference and is often called the "spillover effect". The total treatment

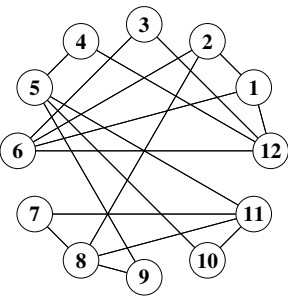 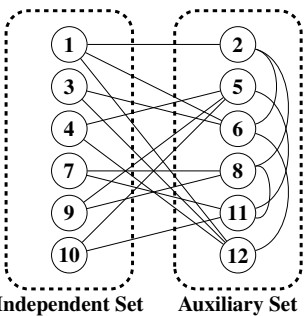 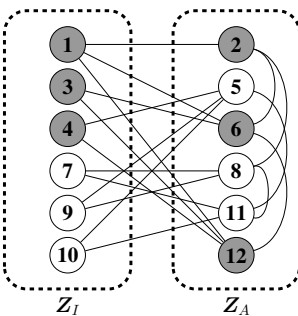

Figure 1: Illustration of Independent Set design. (Left) Example graph. (Middle) The graph is partitioned into an independent set and an auxiliary set. (Right) Conduct an experiment on the independent set and the auxiliary set. Shaded nodes denote the treated nodes.

effect measures the change in outcome when all units are treated vs all units are controlled. Note that the total treatment effect can be decomposed as a combination of direct and indirect effects such that $\tau_i^{(t)} = \tau_i^{(d)}(0) + \tau_i^{(i)}(1,1,0)$.

Experimenters are usually interested in the population-level average effects, which are defined as $\bar{\tau}(z,\rho,z',\rho') := n^{-1}\sum_{i=1}^{n}\tau_i(z,\rho,z',\rho')$. The average direct / spillover / total treatment effect is defined in a similar way. The average total treatment effect $\bar{\tau}^{(t)}$ is of particular interest because it measures the average change in outcomes when all units are assigned to treatment. Further discussions on causal effects under interference can be found in Forastiere et al. (2021).

### 3.2 INDEPENDENT-SET DESIGN

We disentangle the unit-level treatments and the interference by partitioning the units $V$ into two sets, $V_I$ and $V_A$. Let $\mathcal{G}_I = (V_I, E_I)$ be the sub-graph of $\mathcal{G}$ by restricting it to the vertex set $V_I$, and $\mathcal{G}_A = (V_A, E_A)$ be the sub-graph by restricting to $V_A$. Specifically, we require $\mathcal{G}_I$ to be a null graph such that $E_I = \varnothing$, or equivalently, $V_I$ is an **independent set** of $\mathcal{G}$. We call the counterpart $V_A$ as the **auxiliary set**. The middle panel in Figure 1 gives one such partition for the interference graph in the left panel. We will later see that the two sets play different roles in the inference.

Denote the treatment vectors on the independent set $V_I$ and on the auxiliary set $V_A$ by $\boldsymbol{Z}_I$ and $\boldsymbol{Z}_A$, correspondingly. Define the $|V_I| \times |V_A|$ interference matrix $\boldsymbol{\Gamma}$ between $V_I$ and $V_A$ by $[\boldsymbol{\Gamma}]_{ij} := d_i^{-1}\mathbb{I}\{(i,j) \in E\}$, for all $i \in V_I, j \in V_A$, where $d_i$ is the degree of unit $i$ in $\mathcal{G}$, and $\mathbb{I}\{\cdot\}$ is the indicator function. The interference vector on independent set $\boldsymbol{\rho}_I := (\rho_i)_{i \in V_I}$ is given by $\boldsymbol{\rho}_I = \boldsymbol{\Gamma}\boldsymbol{Z}_A$. Because $\boldsymbol{\Gamma}$ is a constant matrix when the graph $\mathcal{G}$ is observed, $\boldsymbol{\rho}_I$ is determined by the auxiliary set's treatment vector $\boldsymbol{Z}_A$. If we restrict our estimation to the independent set $V_I$, the unit-level treatments $\boldsymbol{Z}_I$ and the interference $\boldsymbol{\rho}_I$ can be separately controlled through $\boldsymbol{Z}_I$ and $\boldsymbol{Z}_A$, respectively. We call such a design "independent set design", which partitions the dataset into the independent set and the auxiliary set, controls the treatment and interference separately, and constructs an estimator based on the observed outcomes from the independent set.

The assignments for $\boldsymbol{Z}_I$ and $\boldsymbol{Z}_A$ can be designed to accomplish different causal inference goals. For example, to estimate the average direct effect $\bar{\tau}^{(d)}(\rho)$, one can assign $\boldsymbol{Z}_I$ by completely randomized design and optimize $\boldsymbol{Z}_A$ by minimizing $\|\boldsymbol{\Gamma}\boldsymbol{Z}_A - \rho\|_1$. To estimate the average indirect effect $\bar{\tau}(z,1,0)$, one can assign $Z_i = z$ for all $i \in V_I$ and optimize $\boldsymbol{Z}_A$ by maximizing $\mathrm{Var}[\boldsymbol{\Gamma}\boldsymbol{Z}_A]$. To estimate the average total treatment effect, one can optimize $\boldsymbol{Z}_A$ by maximizing $\mathrm{Var}[\boldsymbol{\Gamma}\boldsymbol{Z}_A]$ and let $Z_i = \mathbf{1}_{\{\rho_i > 0.5\}}$ for all $i \in V_I$. In the right panel of Figure 1, we provide one such assignment of $\boldsymbol{Z}_I$ and $\boldsymbol{Z}_A$ for estimating the total treatment effect. The next sections will discuss the implementations for different causal inference tasks.

Two aspects are compromised to improve the independent set's data quality. The first one is the sample size. Independent set design restricts estimation by using only observations of the independent set, which is usually a fraction of the dataset but of higher quality in estimation. The second aspect that is compromised is the bias from the representativity of the independent set because the

average effect on the independent set is not necessarily identical to the one on the whole dataset. The unbiasedness is achieved when we assume the observed interference graph is random and is unconfounded with the potential outcomes. We discuss the assumptions in Section 4.1.

---

**Algorithm 1:** Greedy algorithm for finding independent set

---

**Input:** graph $\mathcal{G} = (V, E)$
**Output:** independent set $V_I$
$V_I \leftarrow \varnothing$
**while** $|V| > 0$ **do**
  Choose $i$ from $V$ uniformly
  $V_I \leftarrow V_I \cup \{i\}$
  $V \leftarrow V \setminus \{j \in V : (i, j) \in E \text{ or } i = j\}$
**return** $V_I$

---

Before proceeding to different design implementations, we give a greedy algorithm for finding a locally largest independent set in Algorithm 1. The independent set of a graph is not unique in general — any subset of an independent set is another independent set. For inference purposes, one would expect the variance of the estimator scales as $O(|V_I|^{-1})$. Therefore, the goal is to find the largest independent set, which is known as NP-hard (Robson, 1986). In practice, we adopt the fast greedy algorithm that can find the local largest independent set in linear time.

### 3.3 INFERENCE

In this section, we assume the two sets $V_I$ and $V_A$ have been obtained through the aforementioned greedy algorithm and discuss the designs of the assignments on $\boldsymbol{Z}_I$ and $\boldsymbol{Z}_A$ for different inference tasks. For simplicity, we denote their sizes $|V_I|$ and $|V_A|$ by $n_I$ and $n_A$, respectively. Next, we discuss the estimations of the direct treatment effect and spillover effect separately.

#### 3.3.1 ESTIMATING AVERAGE DIRECT EFFECTS

To estimate the average direct effects $\bar{\tau}^{(d)}(\rho)$, one wishes to find an assignment vector $\boldsymbol{Z}_A$ on the auxiliary set such that all independent units' interference received is close to $\rho$. The optimal assignment vector can be found by the following optimization.

$$\min_{\boldsymbol{Z}_A \in \{0,1\}^{n_A}} \|\boldsymbol{\Gamma}\boldsymbol{Z}_A - \rho\boldsymbol{1}\|_1. \tag{2}$$

On the other hand, the treatment vector $\boldsymbol{Z}_I$ for the independent set can be assigned randomly with a completely randomized design such that half of the units are randomly assigned to treated. The difference-in-means estimator on the independent set can be constructed as

$$\hat{\tau}^{(d)}(\rho) = \frac{1}{n_I/2} \sum_{i \in V_I} Y_i^{(obs)} Z_i - \frac{1}{n_I/2} \sum_{i \in V_I} Y_i^{(obs)}(1 - Z_i), \tag{3}$$

where $Y_i^{(obs)}$ is the observed outcome of unit $i$.

The optimization in equation 2 is a convex programming on a binary/integer space with an objective function bounded below by 0. When $\rho = 0$ or $1$, the optimization equation 2 has a trivial but perfect solution as $\boldsymbol{Z}_A = \rho\boldsymbol{1}$ such that all the interference of independent units matches $\rho$ exactly. However, this lower bound is not attainable for general $0 < \rho < 1$. As we will later see in Section 4.3.1, the objective function in equation 2 is directly related to the bias of the estimator in equation 3.

#### 3.3.2 ESTIMATING AVERAGE SPILLOVER EFFECTS AND TOTAL TREATMENT EFFECTS

When estimating the average indirect effect $\bar{\tau}^{(i)}(z, 1, 0)$ or the average total treatment effect $\bar{\tau}^{(t)}$, we hope the interference received by the independent set units spreads to the two extremes: either $\rho_i = 0$ for empty interference or $\rho_i = 1$ for saturated interference. Therefore, we propose to maximize the variance of $(\rho_i)_{i \in V_I}$ in the following optimization:

$$\max_{\boldsymbol{Z}_A \in \{0,1\}^{n_A}} \boldsymbol{Z}_A^T \boldsymbol{\Gamma}^T \left[\boldsymbol{I} - \frac{1}{n_I}\boldsymbol{1}\boldsymbol{1}^T\right] \boldsymbol{\Gamma}\boldsymbol{Z}_A. \tag{4}$$

The above optimization is a concave quadratic programming, and the support can be expanded to its convex hull: $[0, 1]^{n_A}$ while keeping the same solution. The objective function in equation 4 is bounded below by $n_I/4$, the largest eigenvalue of $\boldsymbol{I} - n_I^{-1}\boldsymbol{1}\boldsymbol{1}^T$, by taking $\boldsymbol{\Gamma}\boldsymbol{Z}_A = (1, \ldots, 1, 0, \ldots, 0)$. The optimum may not be attainable when it is outside the manifold of $\boldsymbol{\Gamma}$.

Consider a linear additive structure of the potential outcomes such that

$$Y_i(Z_i, \rho_i) = \alpha + \beta Z_i + \gamma \rho_i + \epsilon_i, \tag{5}$$

where $\epsilon_i$ represents the heterogeneity in potential outcomes and we assume $\text{Var}[\epsilon_i] = \sigma^2$.

With the additive linear model in equation 5, the causal effects are written in a parametric form such that the direct effect is $\beta$, the spillover effect is $\gamma$, and the total treatment effect is $\beta + \gamma$. The coefficients can be estimated through standard linear regression estimators for the units in the independent set. Specifically, the estimators are

$$(\hat{\alpha}, \hat{\beta}, \hat{\gamma})^T = (\boldsymbol{X}^T\boldsymbol{X})^{-1}\boldsymbol{X}^T\boldsymbol{Y}_I,$$

where $\boldsymbol{X} = [\boldsymbol{1}, \boldsymbol{Z}_I, \boldsymbol{\rho}_I]$ is the design matrix and $\boldsymbol{Y}_I$ is the outcome vector on the independent set.

The variance of the estimators $\hat{\beta}$ and $\hat{\gamma}$ depend inversely on the variance of $\boldsymbol{Z}_I$ and $\boldsymbol{\rho}_I$, respectively. The optimization in equation 4 minimizes the variance of linear regression estimators. Detailed discussions on the variance of $\boldsymbol{\rho}_I$ and the variance of the estimator are provided in Section 4.3.2.

Notice that difference-in-means estimators can be used nonparametrically to estimate spillover or total treatment effects. However, this results in fewer units with the required interference levels and, therefore, a large variance.

## 4 THEORETICAL RESULTS

### 4.1 ASSUMPTIONS

We first list a few more assumptions that are necessary for subsequent analysis.

**Assumption 2 (super-population perspective)** *We view the $n$ units from the super-population perspective such that their potential outcomes are i.i.d. from some super-population distribution.*

In Assumption 2, we assume the sample units are i.i.d. from a super-population such that both $V_I$ and $V$ can be viewed as representative finite samples. Furthermore, if we denote $\bar{\tau}_I$ as the average causal effects on the independent set $V_I$, then under Assumption 2, we have $\mathbb{E}[\bar{\tau}_I] = \mathbb{E}[\bar{\tau}]$, which gives the unbiasedness of average causal effect on $V_I$ with respect to the (full) sample average effects.

**Assumption 3 (unconfoundedness of network)** *We assume the observed network $\mathcal{G}$ is a realization of random network, which is unconfounded with the units' potential outcomes such that $\mathcal{G} \perp\!\!\!\perp \mathcal{Y}$, where $\mathcal{Y} := \{Y_i(Z_i, \rho_i) : i \in [n], Z_i \in \{0, 1\}, \rho_i \in [0, 1]\}$ is the table of potential outcomes.*

Furthermore, we assume the network formation is independent of the potential outcomes in Assumption 3. Note that the greedy algorithm of finding the independent set in Algorithm 1, and the optimizations of $\boldsymbol{Z}_A$ as in equation 2 and equation 4 are not node-symmetric — vertices with different degrees have different probabilities of being assigned to $V_I$. It gives rise to the selection bias of the set $V_I$. Assuming the network unconfoundedness, the selection bias from a single sample is averaged if we consider multiple sampling repetitions and network randomization.

### 4.2 THE GREEDY ALGORITHM

The first question at hand is how large the set $V_I$ we can find from a random graph $\mathcal{G}$. Such a graph always exists that one can hardly find a large independent set (for example, the complete graph). Due to the randomness of graph $\mathcal{G}$, we can show such extreme cases occur with a negligible probability as $n \to \infty$. We consider the Erdös-Rényi random graph where each pair of units independently forms an edge with probability $p = s/n$. The random graph has an expected average degree of $pn = s$. Note that most networks of interest (e.g., social networks) are connected and sparse networks. Therefore, we assume $s = o(n)$ for sparsity and assume $s = \Omega(\log n)$ for connectivity. Theorem 1 gives a lower bound on the size of the independent set $V_I$ resulting from Algorithm 1.

**Theorem 1 (lower bound on the greedy algorithm)** *Consider an Erdös-Rényi random graph $\mathcal{G}$ with $n$ vertices and edge probability $p = s/n$ for $s \in \Omega(\log n)$. Then, with high probability, the independent set $V_I$ from the greedy algorithm yields a size at least $\frac{\log s}{s} n$.*

Note that Algorithm 1 is a greedy approximation in finding the locally largest independent set. Dani & Moore (2011) discussed the size of the globally largest independent set for random graphs. The resulting independent set has a similar order of size compared to the globally optimal one. In the egocentric design (Saint-Jacques et al., 2019), the number of ego-clusters is upper bounded by $n/(s+1)$, which is smaller than ours by a factor of $\log s$.

### 4.3 ESTIMATION PERFORMANCE

In this section, we analyze the performance of the causal estimators for the tasks discussed in Section 3.3 and provide additional insights into the optimizations in equation 2 and equation 4.

#### 4.3.1 ESTIMATING AVERAGE DIRECT EFFECT

We first investigate the performance of the difference-in-means estimator in equation 3 for the direct effect. Recall that in order to construct $\hat{\tau}^{(d)}(\rho)$, we first determine $\boldsymbol{Z}_A$ through the optimization in equation 2, and then randomly assign half of $V_I$ to treated through a completely randomized design. Given the sample set, including the potential outcomes $\mathcal{Y}$ and the graph $\mathcal{G}$, for any assignment $\boldsymbol{Z}_A$ on the auxiliary set, we provide an upper bound for the (conditional) bias and variance in Theorem 2.

**Theorem 2 (bias and variance for estimating direct effects)** *Suppose Assumptions 1, 2 and 3 hold, and the potential outcomes $Y_i(z, \rho)$ is uniformly Lipschitz in $\rho$ such that there exists $L > 0$ satisfying $|Y_i(z, \rho_1) - Y_i(z, \rho_2)| \le L|\rho_1 - \rho_2|$, for all $i \in [n], z \in \{0,1\}$ and $\rho_1, \rho_2 \in [0,1]$. Consider the estimator in equation 3. We have*

$$\mathbb{E}[\hat{\tau}^{(d)}(\rho) - \bar{\tau}^{(d)}(\rho) \mid \mathcal{Y}, \mathcal{G}, \boldsymbol{Z}_A] \le \frac{2L}{n_I} \|\boldsymbol{\Delta}\|_1, \quad and$$

$$\left| \mathrm{Var}[\hat{\tau}^{(d)}(\rho) \mid \mathcal{Y}, \mathcal{G}, \boldsymbol{Z}_A] - \frac{1}{n_I} \mathbb{S}_I[Y_i(1, \rho) + Y_i(0, \rho)] \right| \le \frac{4}{n_I(n_I - 1)} \left( LY_{max} \|\boldsymbol{\Delta}\|_1 + L^2 \|\boldsymbol{\Delta}\|_1^2 \right),$$

*where $\boldsymbol{\Delta} = \boldsymbol{\Gamma} \boldsymbol{Z}_A - \rho\boldsymbol{1}$ is the deviation of interference from the target $\rho$, $\mathbb{S}_I[Y_i(1, \rho) + Y_i(0, \rho)] = (n_I - 1)^{-1} \sum_{i \in V_I} (Y_i(1, \rho) + Y_i(0, \rho) - \bar{Y}(1, \rho) - \bar{Y}(0, \rho))^2$ with $\bar{Y}(z, \rho) = n_I^{-1} \sum_{i \in V_I} Y_i(z, \rho)$ is the sample variance of the potential outcomes on $V_I$, and $Y_{max} = \max_{i \in V_I} |Y_i(1, \rho) + Y_i(0, \rho) - \bar{Y}(1, \rho) - \bar{Y}(0, \rho)|$ is the maximum deviation of potential outcomes from their mean on $V_I$.*

$\boldsymbol{\Delta}$ represents the distance between the interference received by the independent units and the target value $\rho$. When $\boldsymbol{\Delta} = \boldsymbol{0}$, the design is equivalent to a completely randomized experiment. In the general case, the optimization in equation 2 minimizes the upper bound for potential bias. Theorem 2 also tells the variance of $\hat{\tau}^{(d)}$ is close to the completely randomized design on $V_I$ if $\|\boldsymbol{\Delta}\|_1$ is small.

#### 4.3.2 ESTIMATING SPILLOVER EFFECT AND TOTAL TREATMENT EFFECT

This section considers the two treatment effects discussed in Section 3.3.2. We focus on the linear additive model in equation 5 such that the effects can be estimated through linear regression. Recall that, for estimating either the spillover effect or the total treatment effect, the assignment $\boldsymbol{Z}_A$ on the auxiliary set is determined through the variance maximization in equation 4. We set $\boldsymbol{Z}_I$ to $z$ for estimating the spillover effect and set $\boldsymbol{Z}_I$ depending on $\boldsymbol{\rho}_I$ for the total treatment effect.

The following theorem gives the bias and variance for the linear regression estimator for the spillover effects. The spillover effect estimator is always unbiased, and its conditional variance depends inversely on the variance of the interference $\boldsymbol{\rho}_I$ of the independent units. The optimization in 4 corresponds to the minimization of the conditional variance of the linear regression estimator.

**Theorem 3 (bias and variance for estimating spillover effects)** *Assume Assumptions 1, 2 and 3. Consider a linear regression over the parametric potential outcome model equation 5 with estimated parameters $\hat{\beta}, \hat{\gamma}$. Set $V_I = z\boldsymbol{1}$ and let $\hat{\tau}^{(i)}(z, 1, 0) = \hat{\gamma}$. Then we have*

$$\mathbb{E}[\hat{\tau}^{(i)}(z, 1, 0) \mid \mathcal{Y}] = \bar{\tau}^{(i)}(z, 1, 0) \quad and \quad \mathrm{Var}[\hat{\tau}^{(i)}(z, 1, 0) \mid \mathcal{Y}, \mathcal{G}, \boldsymbol{Z}_A] = \frac{\sigma^2}{n_I \mathrm{Var}[\boldsymbol{\rho}_I]}.$$

Theorem 4 gives the unbiasedness of the total treatment effect estimator and provides a lower bound on the variance. Such a lower bound is, in general, not attainable in the binary support of $\boldsymbol{Z}_A$, but it provides the reciprocal dependence on $\mathrm{Var}[\boldsymbol{\rho}_I]$, which was maximized in the optimization 4 as well.

**Theorem 4 (bias and variance for estimating total treatment effects)** *Assume Assumptions 1, 2 and 3. Consider a linear regression over the parametric potential outcome model equation 5 with estimated parameters $\hat{\beta}, \hat{\gamma}$. Let $\hat{\tau}^{(t)} = \hat{\beta} + \hat{\gamma}$. If $|\mathrm{Corr}(\boldsymbol{Z}_I, \boldsymbol{\rho}_I)| < 1$, we have*

$$\mathbb{E}[\hat{\tau}^{(t)} \mid \mathcal{Y}] = \bar{\tau}^{(t)}, \quad and \quad \mathrm{Var}[\hat{\tau}^{(t)} \mid \mathcal{G}, \boldsymbol{Z}_A, \boldsymbol{Z}_I] = \frac{\sigma^2}{n_I} \frac{\mathrm{Var}[\boldsymbol{Z}_I - \boldsymbol{\rho}_I]}{\mathrm{Var}[\boldsymbol{Z}_I]\mathrm{Var}[\boldsymbol{\rho}_I] - \mathrm{Cov}^2[\boldsymbol{Z}_I, \boldsymbol{\rho}_I]}.$$

*The variance is lower bounded by $\sigma^2/(n_I \mathrm{Var}[\boldsymbol{\rho}_I])$.*

## 5 EXPERIMENTS

In this section, we implement the independent set design and validate results empirically on synthetic graphs. In order to evaluate the performance of the proposed design, we conduct simulations under a variety of simulated settings. The bias and variance of estimators are compared for completely randomized design on the independent set (CR), completely randomized design on the full graph (Full), ego-clusters, graph cluster, and Independent Set design (IS), which optimizes the assignments on the auxiliary set. To illustrate the benefits of IS design, we implement the design to estimate average spillover effects and average direct effects, respectively.

### 5.1 ESTIMATING AVERAGE SPILLOVER EFFECTS

First, we examine the performance of estimating average spillover effects for different designs. We run experiments on 50 Erdös-Rényi random graphs (Erdős & Rényi, 1959) with $n = 60, p = 0.1$, and estimate the spillover effects. The potential outcomes are randomly generated by $Y_i(Z_i, \rho_i) = \alpha + \beta Z_i + \gamma \rho_i + U + \epsilon_i, \forall i \in V$, and we let baseline $\alpha = 1$, treatment parameter $\beta = 20$, interference parameter $\gamma = 5, 10, 15, 20, U \sim \mathrm{Unif}(0, 1)$ and $\epsilon_i \sim N(0, 0.5)$.

Results are displayed in Figure 2. The independent-set design outperforms all other methods in estimating the average spillover effects. At different levels of interference, IS design shows the lowest bias and variance. We notice that the IS design is more stable than the other methods, and the variance of the IS design stabilizes at a very low level as spillover effects increase.

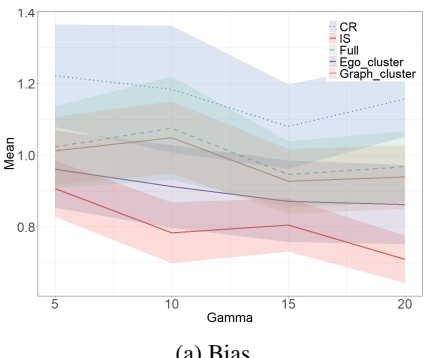
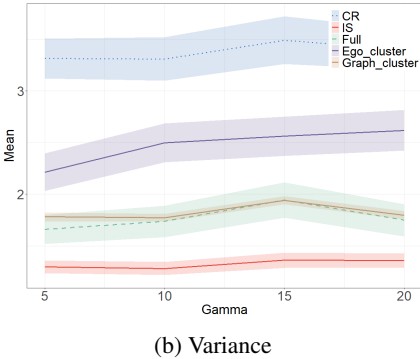

(a) Bias

(b) Variance

Figure 2: Bias and variance to estimate the average spillover effects. The bands around the lines represent the errors of the estimator for each value of Gamma.

To introduce diversity, we run experiments on distinct random graphs: Erdös-Rényi (Erdős & Rényi, 1959) $G(n, p)$, Barabási–Albert (Barabási & Albert, 1999) $G(n, m)$ and small-world (Watts & Strogatz, 1998) $G(n, p)$. The potential outcomes are based on $Y_i(Z_i, \rho_i) = \alpha_i + \beta Z_i + \gamma \rho_i + \epsilon_i, \forall i \in V$, and $\alpha_i = 1, \beta = 20, \gamma = 10$. $\epsilon_i \sim N(0, 0.5)$. Each configuration is run 2,000 times. The results of estimating spillover effects are summarized in Tables 2. As shown in Tables 2, IS design outperforms all other methods on distinct settings and achieves the lowest bias and variance.

Table 2: Performance in estimating average spillover effects under distinct random graphs.

| Graph | Parameters | CR | | IS | | Full | | Graph Cluster | | Ego-Clusters | |
|---|---|---|---|---|---|---|---|---|---|---|---|
| | | Bias | Variance | Bias | Variance | Bias | Variance | Bias | Variance | Bias | Variance |
| Erdös-Rényi | n = 100, p = 0.10 | 0.598 | 0.468 | 0.398 | 0.242 | 0.473 | 0.399 | 0.497 | 0.418 | 0.428 | 0.491 |
| | n = 200, p = 0.15 | 0.392 | 0.188 | 0.315 | 0.124 | 0.366 | 0.178 | 0.384 | 0.192 | 0.342 | 0.187 |
| | n = 400, p = 0.15 | 0.246 | 0.096 | 0.225 | 0.067 | 0.266 | 0.094 | 0.242 | 0.089 | 0.239 | 0.091 |
| Barabási–Albert | n = 100, m = 1 | 0.192 | 0.064 | 0.152 | 0.032 | 0.168 | 0.049 | 0.177 | 0.056 | 0.159 | 0.072 |
| | n = 75, m = 1 | 0.239 | 0.055 | 0.135 | 0.041 | 0.181 | 0.051 | 0.185 | 0.067 | 0.176 | 0.079 |
| Small world | n = 80, p = 0.05 | 0.303 | 0.165 | 0.212 | 0.087 | 0.263 | 0.089 | 0.274 | 0.093 | 0.232 | 0.117 |
| | n = 50, p = 0.05 | 0.351 | 0.093 | 0.243 | 0.036 | 0.277 | 0.044 | 0.296 | 0.061 | 0.264 | 0.089 |

## 5.2 ESTIMATING AVERAGE DIRECT EFFECTS

We will next implement an IS design to estimate average direct effects and evaluate performance under various settings. Here, the potential outcomes are randomly generated according to $Y_i(Z_i, \rho_i) = \alpha_i + \beta Z_i + \gamma \rho_i + \epsilon_i, \ \forall i \in V$, and we have $\alpha_i = 1, \beta = 20, \gamma = 10$. $\epsilon_i \sim N(0, 0.5)$. We run experiments 2,000 times and record the bias and variance of estimators. Table 3 shows the results of estimating average direct effects. In the presence of spillover effects, IS design achieves the best performance (the lowest bias and variance). When SUTVA is violated, network interference impairs the performance of conventional approaches. In this case, estimators in classical designs are biased in estimating average direct effects. In contrast, IS design could mitigate the spillover effects and improve performance by controlling treatment assignments on the auxiliary set.

Table 3: Performance of designs in estimating average direct effects under distinct random graphs.

| Graph | Parameters | CR | | IS | | Full | | Graph Cluster | |
|---|---|---|---|---|---|---|---|---|---|
| | | Bias | Variance | Bias | Variance | Bias | Variance | Bias | Variance |
| Erdös-Rényi | n = 100, p = 0.1 | 0.262 | 0.035 | 0.117 | 0.009 | 0.261 | 0.023 | 0.221 | 0.024 |
| | n = 200, p = 0.1 | 0.193 | 0.038 | 0.094 | 0.005 | 0.188 | 0.015 | 0.156 | 0.013 |
| | n = 400, p = 0.1 | 0.098 | 0.006 | 0.062 | 0.002 | 0.086 | 0.004 | 0.081 | 0.005 |
| Barabási–Albert | n = 80, m = 1 | 1.145 | 0.851 | 0.237 | 0.034 | 1.130 | 0.949 | 1.031 | 0.815 |
| | n = 100, m = 1 | 1.207 | 0.693 | 0.278 | 0.026 | 1.296 | 0.832 | 1.139 | 0.742 |
| Small world | n = 100, p = 0.05 | 0.961 | 0.556 | 0.278 | 0.024 | 1.184 | 0.791 | 1.105 | 0.716 |
| | n = 50, p = 0.05 | 1.101 | 0.869 | 0.225 | 0.029 | 1.115 | 0.877 | 1.013 | 0.854 |

## 6 DISCUSSION AND CONCLUSION

Randomization experiments on networks can be biased due to interference among units, making causal inference on units challenging. In this study, we propose a new randomized design that separates units into independent and auxiliary sets to control interference. Our design can be applied to any network and provides accurate estimations with good interpretability.

Whilst IS design allows us to improve the estimation of causal effects on networks, there are several limitations. Due to the segmentation of units in a network, we can only estimate causal effects on the independent set. The experiment's sample size will be smaller than using the full network. Moreover, the computational cost depends on the size of the auxiliary set, so it may take more time to optimize the assignments on the auxiliary set. Another limitation is that our design depends on observed networks. The performance of the proposed design on a misspecified network is unknown. Future research includes improving the computational efficiency of the algorithm to optimize the assignments on the auxiliary set and further extending it when the network is misspecified.

## ACKNOWLEDGEMENTS

This work was supported in part by NSF award IIS-1409177 and by ONR award N00014-17-1-2131.

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

# Appendix

## A    PROOF OF THEOREM 1

In order to prove the lower bound, we follow the differential equation method in (Frieze & Karoński, 2016, Chapter 24).

Recall Algorithm 1, and let $v_t, t = 1, \ldots, |V_I|$ be the selected independent unit at step $t$. Denote $V(t)$ the set of vertices after step $t$ in the algorithm such that $V(0) = n$ and $V(|V_I|) = 0$. Write $X(t) = |V(t)|$ as the number of remaining units after picking out $t$ independent units. Then the algorithm gives

$$X(t+1) = X(t) - 1 - d(v_{t+1} \mid V(t)), \text{ for } t = 0, \ldots, |V_I| - 1,$$

where $d(v_{t+1} \mid V(t))$ is the degree of $v_{t+1}$ in the subgraph containing the vertex set $V(t)$. Let $f(t, x) = -sx - (1 - p)$ and

$$D = \left\{ (t, x) : -\frac{1}{n} < t < 1, 0 < x < 1 \right\}.$$

We now show the conditions (P1)-(P4) in (Frieze & Karoński, 2016, Chapter 24) are satisfied.

**(P1)**: It is obvious $X(t) < X(0) = n$. Therefore, (P1) is satisfied with $C_0 = 1$.
**(P2)**: Since $|X(t+1) - X(t)| = d(v_{t+1} \mid V(t)) + 1 \le d(v_{t+1} \mid V_0) + 1$, where $v_t$ is the vertex deleted at step $t$. We claim with high probability,

$$d(v_{t+1} \mid V_0) \le \max_{v \in G} d(v \mid V_0) \le 2(s + \log n) - 1.$$

It follows from

$$\mathbb{P} \left[ \max_{v \in G} d(v \mid V_0) \ge \xi \right] \le \sum_{v \in G} \mathbb{P}[d(v \mid V_0) \ge \xi]$$

$$\overset{(i)}{\le} n \exp \left\{ -\frac{n-1}{n} s \left( \frac{n\xi}{(n-1)s} \log \frac{n\xi}{(n-1)s} - \frac{n\xi}{(n-1)s} + 1 \right) \right\}$$

$$\le \exp \left\{ \log n - \xi \log \frac{\xi}{s} + \xi \right\},$$

where $(i)$ follows from the Chernoff bound of binomial distributions (see, for example, Chapter 2 in Wainwright (2019)).
By setting $\xi = e^2(s + \log n) - 1 > e^2 s$, we have

$$\mathbb{P} \left[ \max_{v \in G} d(v \mid V_0) \ge e^2(s + \log n) - 1 \right] \le n e^{-\xi} \le \frac{e}{n^{e^2 - 1}} e^{-e^2 s}.$$

Therefore,

$$|X(t+1) - X(t)| \le e^2(s + \log n)$$

is satisfied with probability at least $1 - n^{1-e^2} e^{1-e^2 s} \ge 1 - n^{-6} e^{-6s}$.
**(P3)**: Let $\mathcal{E}$ be the event that (P2) holds. Then,

$$|\mathbb{E}(X(t+1) - X(t) \mid V(t), \mathcal{E}) - f(t/n, X(t)/n)| = 0.$$

**(P4)**: It is immediate that $|f(t, x) - f(t', x')| = s|x - x'|$.

To use Theorem 24.1 in Frieze & Karoński (2016), we set

$$C_0 = 1$$
$$\beta = e^2(s + \log n)$$
$$\lambda = n^{\epsilon - 1/3}(s + \log n)$$
$$\gamma = n^{-6} e^{-6s}$$
$$\alpha = n^{3\epsilon} e^{-6}.$$

Then we have $X(t) = nz(t/n) + O(n^{\epsilon+2/3}(s + \log n))$ uniformly on $0 \le t \le \sigma t$, with probability at least $1 - O(n^{-6}e^{-6s} + n^{1/3-\epsilon}e^{-n^{3\epsilon}e^{-6}}) = 1 - O(n^{-6}e^{-6s}) \ge 1 - O(n^{-12})$.

$z(t)$ satisfies the following ODE that:

$$z'(t) = f(t, z(t)) = -sz(t) - (1 - p),$$

with initial condition that $z(0) = X(0) = n$. The solution is

$$z(t) = -\frac{1-p}{s} + \left(1 + \frac{1-p}{s}\right)e^{-st}.$$

We have $z(t) > 0$ on $\left[0, \frac{1}{s}\log\frac{s+1-p}{1-p}\right] \supset \left[0, \frac{\log s}{s}\right]$. Therefore, with high probability, $X(t) \ge 0$ for $t \le \frac{\log s}{s}n$.

In conclusion, we have at least $\frac{\log s}{s}n$ iterations in the greedy algorithm, resulting in an independent set of size at least $\frac{\log s}{s}n$ with probability at least $1 - O(n^{-12})$.

## B   PROOF OF THEOREM 2

We fix the potential outcomes $\mathcal{Y}$, the graph $\mathcal{G}$, and the assignment $\mathbf{Z}_A$ for the auxiliary set. Recall the difference-in-means estimator:

$$\hat{\tau}^{(d)}(\rho) = \frac{1}{n_I/2}\sum_{i \in V_I} Y_i^{(obs)}Z_i - \frac{1}{n_I/2}\sum_{i \in V_I} Y_i^{(obs)}(1 - Z_i).$$

We have the expectation:

$$\mathbb{E}[\hat{\tau}^{(d)}(\rho) \mid \mathcal{Y}, \mathcal{G}, \mathbf{Z}_A] = \frac{1}{n_I}\sum_{i \in V_I} Y_i(1, \rho_i) - \frac{1}{n_I}\sum_{i \in V_I} Y_i(0, \rho_i),$$

where we use the facts that $\rho_i$ is fixed given $\mathcal{G}$ and $\mathbf{Z}_A$, and $\mathbb{E}[Z_i] = 1/2$. Therefore, the bias is

$$\left|\mathbb{E}[\hat{\tau}^{(d)}(\rho) \mid \mathcal{Y}, \mathcal{G}, \mathbf{Z}_A] - \bar{\tau}^{(d)}(\rho)\right| = \left|\frac{1}{n_I}\sum_{i \in V_I}[Y_i(1, \rho_i) - Y_i(1, \rho)] - \frac{1}{n_I}\sum_{i \in V_I}[Y_i(0, \rho_i) - Y_i(1, \rho)]\right|$$

$$\le \frac{2L}{n_I}\sum_{i \in V_i}|\rho_i - \rho|$$

$$= \frac{2L}{n_I}\|\boldsymbol{\rho}_I - \rho\mathbf{1}\|_1.$$

Next, we give the variance.

$$\mathrm{Var}[\hat{\tau}^{(d)}(\rho) \mid \mathcal{Y}, \mathcal{G}, \mathbf{Z}_A]$$

$$= \mathrm{Var}\left[\frac{1}{n_I/2}\sum_{i \in V_I}[Y_i(1, \rho_i) + Y_i(0, \rho_i)]Z_i\right]$$

$$= \frac{4}{n_I^2}\sum_{i \in V_I}[Y_i(1, \rho_i) + Y_i(0, \rho_i)]^2\mathrm{Var}[Z_i]$$

$$\quad + \frac{4}{n_I^2}\sum_{i \ne j \in V_I}[Y_i(1, \rho_i) + Y_i(0, \rho_i)][Y_j(1, \rho_j) + Y_j(0, \rho_j)]\mathrm{Cov}[Z_i, Z_j]$$

$$= \frac{1}{n_I^2}\sum_{i \in V_I}[Y_i(1, \rho_i) + Y_i(0, \rho_i)]^2 - \frac{1}{n_I^2(n_I - 1)}\sum_{i \ne j \in V_I}[Y_i(1, \rho_i) + Y_i(0, \rho_i)][Y_j(1, \rho_j) + Y_j(0, \rho_j)]$$

$$= \frac{1}{n_I}\mathbb{S}_I[Y_i(1, \rho_i) + Y_i(0, \rho_i)]$$

Therefore, we have

$$
\left| \mathrm{Var}[\hat{\tau}^{(d)}(\rho) \mid \mathcal{Y}, \mathcal{G}, \boldsymbol{Z}_A] - \frac{1}{n_I} \mathbb{S}_I[Y_i(1, \rho) + Y_i(0, \rho)] \right|
$$

$$
= \frac{1}{n_I} \left[ 2\mathbb{S}_I[Y_i(1, \rho_i) + Y_i(0, \rho_i)] - \mathbb{S}_I[Y_i(1, \rho) + Y_i(0, \rho)]] \right]
$$

$$
= \frac{1}{n_I} \left( 2\mathbb{S}_I[\delta_i, Y_i(1, \rho) + Y_i(0, \rho)] + \mathbb{S}_I[\delta_i] \right)
$$

$$
\leq \frac{1}{n_I(n_I - 1)} \left[ 2\|\boldsymbol{\delta}_I\|_1 Y_{max} + \|\boldsymbol{\delta}_I\|_2^2 \right]
$$

$$
\leq \frac{4}{n_I(n_I - 1)} \left[ L\|\boldsymbol{\Delta}_I\|_1 Y_{max} + L^2\|\boldsymbol{\Delta}_I\|_1^2 \right],
$$

where $\delta_i = Y_i(1, \rho_i) + Y_i(0, \rho_i) - Y_i(1, \rho) + Y_i(0, \rho) \leq 2L\Delta_i$, and

$$
Y_{max} = \max_{i \in V_i} \left| Y_i(1, \rho) + Y_i(0, \rho) - \frac{1}{n_I} \sum_{i \in V_I} [Y_i(1, \rho) + Y_i(0, \rho)] \right|.
$$

.

## C  PROOF OF THEOREM 3

Consider the linear regression $Y_i \sim \alpha + \beta Z_i + \gamma \rho_i$ for $i \in V_I$. When $Z_i$ is a constant as in the design, the linear regression is equivalent to $Y_i \sim \tilde{\alpha} + \gamma \rho_i$, where $\tilde{\alpha} := \alpha + \beta z$. By observing $\hat{\tau}^{(i)}(z, 1, 0) = \hat{\gamma}$, the result follows immediately from univariate linear regression.

## D  PROOF OF THEOREM 4

Let $\boldsymbol{Y}_I$ be the observed outcomes on the independent set such that $\boldsymbol{Y}_I = \alpha \mathbf{1} + \beta \boldsymbol{Z}_I + \gamma \boldsymbol{\rho}_I + \boldsymbol{\epsilon}_I$. We write the design matrix as

$$
\boldsymbol{X} = \begin{bmatrix} \mathbf{1} & \boldsymbol{Z}_I & \boldsymbol{\rho}_I \end{bmatrix} = \begin{bmatrix} \mathbf{1} & \tilde{\boldsymbol{Z}}_I & \tilde{\boldsymbol{\rho}}_I \end{bmatrix} \begin{bmatrix} 1 & \bar{Z}_I & \bar{\rho}_I \\ 0 & 1 & 0 \\ 0 & 0 & 1 \end{bmatrix},
$$

where $\bar{Z}_I$ and $\bar{\rho}_I$ are sample means of $\boldsymbol{Z}_I$ and $\boldsymbol{\rho}_I$, correspondingly, and $\tilde{\boldsymbol{Z}}_I$ and $\tilde{\boldsymbol{\rho}}_I$ are de-meaned versions of $\boldsymbol{Z}_I$ and $\boldsymbol{\rho}_I$, correspondingly.

Let $\hat{\alpha}$, $\hat{\beta}$, $\hat{\gamma}$ be the regression estimators. The unbiasedness of $\hat{\beta} + \hat{\gamma}$ to the true value $\beta + \gamma$ is immediate. Now we calculate the covariance matrix such that

$$
\mathrm{Cov}[(\hat{\alpha}, \hat{\beta}, \hat{\gamma})] = \sigma^2 [\boldsymbol{X}^T \boldsymbol{X}]^{-1}
$$

$$
= \sigma^2 \begin{bmatrix} 1 & -\bar{Z}_I & -\bar{\rho}_I \\ 0 & 1 & 0 \\ 0 & 0 & 1 \end{bmatrix} \begin{bmatrix} n_I & 0 & 0 \\ 0 & n_I \mathrm{Var}[\boldsymbol{Z}_I] & n_I \mathrm{Cov}[\boldsymbol{Z}_I, \boldsymbol{\rho}_I] \\ 0 & n_I \mathrm{Cov}[\boldsymbol{Z}_I, \boldsymbol{\rho}_I] & n_I \mathrm{Var}[\boldsymbol{\rho}_I] \end{bmatrix}^{-1} \begin{bmatrix} 1 & 0 & 0 \\ -\bar{Z}_I & 1 & 0 \\ -\bar{\rho}_I & 0 & 1 \end{bmatrix}
$$

$$
= \frac{\sigma^2}{n_I} \begin{bmatrix} 1 & -S^{-1}(\overline{Z}_I \overline{\rho^2} - \overline{\rho}\overline{Z_I \rho}) & S^{-1}(\overline{Z}_I \overline{Z_I \rho} - \overline{\rho}\overline{Z^2}) \\ -S^{-1}(\overline{Z}_I \overline{\rho^2} - \overline{\rho}\overline{Z_I \rho}) & S^{-1}\mathrm{Var}[\boldsymbol{\rho}_I] & -S^{-1}\mathrm{Cov}[\boldsymbol{Z}_I, \boldsymbol{\rho}_I] \\ S^{-1}(\overline{Z}_I \overline{Z_I \rho} - \overline{\rho}\overline{Z^2}) & -S^{-1}\mathrm{Cov}[\boldsymbol{Z}_I, \boldsymbol{\rho}_I] & S^{-1}\mathrm{Var}[\boldsymbol{Z}_I] \end{bmatrix},
$$

where $S = \mathrm{Var}[\boldsymbol{Z}_I]\mathrm{Var}[\boldsymbol{\rho}_I] - \mathrm{Cov}^2[\boldsymbol{Z}_I, \boldsymbol{\rho}_I]$.
Therefore, we have

$$
\mathrm{Var}[\hat{\beta} + \hat{\gamma}] = \frac{\sigma^2}{n_I} \frac{\mathrm{Var}[\boldsymbol{Z}_I] + \mathrm{Var}[\boldsymbol{\rho}_I] - 2\mathrm{Cov}[\boldsymbol{Z}_I, \boldsymbol{\rho}_I]}{\mathrm{Var}[\boldsymbol{Z}_I]\mathrm{Var}[\boldsymbol{\rho}_I] - \mathrm{Cov}^2[\boldsymbol{Z}_I, \boldsymbol{\rho}_I]}.
$$

Consider the function $f(\lambda)$:

$$
\lambda \mapsto \frac{\mathrm{Var}[\boldsymbol{Z}_I] + \mathrm{Var}[\boldsymbol{\rho}_I] - 2\lambda}{\mathrm{Var}[\boldsymbol{Z}_I]\mathrm{Var}[\boldsymbol{\rho}_I] - \lambda^2}.
$$

Then we have

$$\frac{df}{d\lambda} = -\frac{2(\lambda - \mathrm{Var}[\boldsymbol{Z}_I])(\lambda - \mathrm{Var}[\boldsymbol{\rho}_I])}{(\mathrm{Var}[\boldsymbol{Z}_I]\mathrm{Var}[\boldsymbol{\rho}_I] - \mathrm{Cov}^2[\boldsymbol{Z}_I, \boldsymbol{\rho}_I])^2}.$$

Therefore, when $\lambda^2 < \mathrm{Var}[\boldsymbol{Z}_I]\mathrm{Var}[\boldsymbol{\rho}_I]$, $f(\lambda)$ takes its minimum at $\lambda^* = \mathrm{Var}[\boldsymbol{Z}_I] \wedge \mathrm{Var}[\boldsymbol{\rho}_I]$ such that

$$f(\lambda^*) = \frac{1}{\mathrm{Var}[\boldsymbol{Z}_I] \wedge \mathrm{Var}[\boldsymbol{\rho}_I]} \geq \frac{1}{\mathrm{Var}[\boldsymbol{\rho}_I]}.$$

Therefore, we have

$$\mathrm{Var}[\hat{\beta} + \hat{\gamma}] \geq \frac{\sigma^2}{n_I \mathrm{Var}[\boldsymbol{\rho}_I]}.$$

The lower bound can be obtained by choosing $\boldsymbol{Z}_I = \boldsymbol{\rho}_I \in [0,1]^{n_I}$, generally outside the binary support for $\boldsymbol{Z}_I$.

