# OpenReview forum: "Independent-Set Design of Experiments for Estimating Treatment and Spillover Effects under Network Interference"
_ICLR.cc/2024/Conference — ICLR 2024 poster_

### Official Review · Reviewer_hHf8 · 2023-10-22

**Soundness:** 4 excellent
**Presentation:** 2 fair
**Contribution:** 2 fair
**Rating:** 5
**Confidence:** 4

**Summary:**

Interference is a common problem in experimental designs that biases estimation of treatment effects. This paper attempts to correct for interference by designing an experiment on a subset of the data that consists of non-interfering units. Given an interference network this paper provides an algorithm for treating units for various parameter goals. The paper provides a comparison on the sample efficiency of the method related to other methods and provides bias and variance computations.

**Strengths:**

- The paper provides a novel algorithm for treating units to minimize the effects of interference and maximizing sample efficiency for specific graph classes.
- It is also good that the paper provides an optimization framework that can be used for computation of various causal parameters. This is something that is often missing in interference papers.
- Theoretical analysis provides proof of unbiasedness and computation of estimator variance showcasing theoretical proof of method.
- While the theoretical section relies on some stringent assumptions, i.e. G \perp Y, the algorithm itself is relatively assumption free aside from a requirement of sparsity which is often true in practice

**Weaknesses:**

It is difficult to assess the contribution of the paper because the idea of designing experiments on non-interfering units has been well studied in previous works -- this is documented in the paper's related works. This paper attempts to design a more robust methodology towards this idea but relies on knowledge of the underlying network G. In practice this is never known and the paper does not consider the case of a misspecified G.

The theoretical results are good but they rely on stringent assumptions (although possibly weaker than other works). In particular it seems that the sample efficiency results are specific to the case of an erdos-renyi graph. Furthermore, the methodology rules out complete graphs and in general requires a degree of sparsity in the underlying graph. Since G is given or estimated this could be enforced but possibly unpalatable in some cases. Furthermore, as with other approaches to interference using partitioning algorithm there is a trade off with statistical power.

**Questions:**

- How does this method work when G is misspecified?
- It would be good to see how this could be applied in a practical example with different underlying graphs
- How does optimizing over treatments effect the randomization assumption? Could this effect the internal validity of the study if randomization is weakened (i.e. in the vein of Bugni, Canay, and Shaikh 2017) moreover could there be distributional differences between the independent set and aux sets?
- Since treatment effects can only be computed on the independent set how does this impact possible external validity of the study?

---

> ### Author Response · Authors · 2023-11-23
>
> We thank the reviewer for the careful comments on our manuscript. For the weakness part, the reviewer commented on the few assumptions that are proposed in the paper. We list our responses as below:
> 1. In our paper, we assume the network is observed at least before the partitioning step. This is usually the case. For example, the interference can the friendship network in social networks or the transportation network for the supply chain problems. It is a common assumption (see Ugander et al., 2013; Forastiere et at., 2021; Cai et al., 2022)  Our work focuses on observed network, and the design of experiment is on observed networks, but it is a good direction for the future work when network G is misspecified,  and we include that in our discussion as future work.
>  2. We agree it is always helpful to inspect models on real data. However, the paper focus more on the design and theory— while in real data we never know the true potential outcomes and true effects. Therefore, we compare our design with other designs on the synthetic datasets, where we can artificially set the truth and compare the performances.
>  3. If we understand the reviewer’s question correctly, it is about the representativeness of the outcomes after a deterministic assignment of the treatments. We admit that, for a given sample set (that is a fixed potential outcomes and a fixed observed network), the greedy algorithm and the optimization of assignments do indeed weaken the representativeness of the observed outcomes because the vertices has unequal probabilities of being assigned to treated or being partitioned to the independent set. This is why we are NOT claiming our estimation is unbiased for the given sample set of data. Instead, we DO claim the unbiasedness from the super-population perspective (Assumption 2), where all samples along with their potential outcomes and their network are assumed to be sampled from the population. We would refer to Section 7.3 of Imbens and Rubin (2015) for the super-population framework. This is related to our response to point 9 of reviewer DrFU.
> 4. See the response to question 3
>
> Reference:
>
> - Johan Ugander, Brian Karrer, Lars Backstrom, and Jon Kleinberg. Graph cluster randomization: Network exposure to multiple universes. In Proceedings of the 19th ACM SIGKDD International Conference on Knowledge Discovery and Data Mining, KDD ’13, pp. 329–337, 2013.
> - Laura Forastiere, Edoardo M Airoldi, and Fabrizia Mealli. Identification and estimation of treatment and interference effects in observational studies on networks. Journal of the American Statistical Association, 116(534):901–918, 2021.
> - Chencheng Cai, Jean Pouget-Abadie, and Edoardo M Airoldi. Optimizing randomized and deterministic saturation designs under interference. arXiv preprint arXiv:2203.09682, 2022.
> - Imbens, G. W. and Rubin, D. B., Causal Inference for Statistics, Social, and
> Biomedical Sciences: An Introduction, Cambridge University Press, 2015.

---

### Official Review · Reviewer_DrFU · 2023-10-31

**Soundness:** 3 good
**Presentation:** 3 good
**Contribution:** 3 good
**Rating:** 6
**Confidence:** 3

**Summary:**

The paper proposes experimental designs using the independent set approach to estimate both direct and spillover effects for general networks. Both theoretical justification and experimental results are provided to demonstrate advantages over competing designs.

**Strengths:**

Many interference designs focus on estimating the total effect, whereas the literature focusing on designs to measure the spillover effect specifically is more limited. This is an important problem and the paper proposes optimal designs that are flexible and also show strong theoretical and experimental results.

The proposed designs are novel and simple to implement.

**Weaknesses:**

The experimentation section could use more polish/clarity, and possibly some additional exploration (see questions below). In particular, there are inconsistencies in the displayed results and limitations due to the greedy algorithm, as well as assumption 2, could be better addressed in the experiments.

**Questions:**

Experimentation Questions:
- Inconsistency: Why does Figure 2 use n=60 whereas Table 2 uses n={100,200,400}?
- Why is Graph Cluster omitted from Figure 2?
- Graph Cluster is not mentioned in the Section 5 introduction, what cluster design is used?
- Is there a reason some designs mentioned in Section 2 (ego-clusters, randomized saturation) appear to be omitted?
- Given the discussion in Section 4.2, how do results vary for more varied p/s specifications for ER random graphs?
- How robust are the results to the size of the independent set?
- What is the $\rho$ chosen for the IS design in section 5.2? Is it the usual IS setting from Karwa and Airoldi where $\rho = 0$?

Minor nomenclature question: The direct effect should be $\tau_{i}^{(d)}(0)$ and $\tau_{i}^{(d)}(\rho)$ represents the total effect in a partial spillover situation, correct?

How reasonable is Assumption 2 given the greedy algorithm to construct $V_I$ has no concern for representativeness?

---

> ### Author Response · Authors · 2023-11-23
>
> We are most grateful for the constructive comments from the reviewer on our manuscript. For the weakness part, we would like to clarify the questions the reviewer commented on in a point-to-point fashion.
> 1. Figure 2 illustrates a detailed visualization of confidence intervals to show how bias and variance for different designs change when interference gamma increases. We use n = 60 as an illustration example. Table 2 provides more comparisons of different designs on distinct random graphs when interference is fixed, hence we let n = {100, 200, 400} to introduce diversity.
> 2. We add the simulation results of  graph cluster randomization to the revised manuscript in Figure 2.
> 3. Graph clustering is same as the randomized saturation design except that the user needs to cluster the units when no clear cut-offs exist. We use multi-level modularity optimization(Blondel, Vincent D., et al, 2008) for the clustering. Since there is no clear cluster boundary in the network, graph clustering is inferior to the independent set design due to significant bias.To better illustrate the performance of graph clustering,  we add the simulation results of graph cluster randomization to the revised manuscript.
>  4. Some of the designs listed in Table 1 are widely used designs, which were not necessarily designed for the causal inference over a well-connected interference network. The performance of other designs in Table 1 are obviously inferior to our approach. That is why only a few methods are compared for demonstration purpose. Additional results of comparison with other designs in Table 1 have included in the revised manuscript, include ego-clusters and graph cluster (similar to randomized saturation).
>  5. Thm 1 from Section 4.2 provides a lower bound for the size of the independent set, which suffices to show its superiority over other methods. In the simulation part, the variance roughly scales as 1/n, which is much better than the lower bound provided in Section 4.2.
> 6. The variance scales as 1/n_I from the theoretical perspective.
> 7. in 5.2 we let \rho = 0 when estimating  ATE(the average direct effects)
> 8. For the nomenclature question, yes, they can be considered as the direct effect and the total effect.
> 9. In practice, any sub-sample/separation design involves the representativeness issue on a fixed sample dataset. On the one hand, the first step in the greedy algorithm involves a random choice of the first vertex. Every vertex has a positive probability of being selected to the independent set. The bias from the greedy algorithm is alleviated by the stochastic algorithm, similar to previous work, such as Saint-Jacqueset al., 2019;  Uganderetal.,2013.  On the other hand, we are aiming to estimate the population causal effect instead of the sample averaged causal effect. A bias in the representativeness of the independent set to the sample set is mitigated by considering the super-population perspective, where, under repeated experiments, the network and the potential outcome are randomly drawn from the population. Therefore, the estimator is unbiased for the populational causal effect in the super-population framework (instead of the unbiasedness for sample average effect).
>
> Reference:
> -  Blondel, V. D., Guillaume, J. L., Lambiotte, R., & Lefebvre, E. (2008). Fast unfolding of communities in large networks. Journal of Statistical Mechanics: Theory and Experiment, 2008(10), P10008.
> -  Imbens, G. W. and Rubin, D. B., Causal Inference for Statistics, Social, and
> Biomedical Sciences: An Introduction, Cambridge University Press, 2015.
> - Guillaume Saint-Jacques, Maneesh Varshney, Jeremy Simpson, and Ya Xu. Using ego-clusters to measure network effects at linkedin. arXiv preprint arXiv:1903.08755, 2019.
> - Johan Ugander, Brian Karrer, Lars Backstrom, and Jon Kleinberg. Graph cluster randomization: Network exposure to multiple universes. In Proceedings of the 19th ACM SIGKDD International Conference on Knowledge Discovery and Data Mining, KDD ’13, pp. 329–337, 2013.

---

### Official Review · Reviewer_vRUq · 2023-10-31

**Soundness:** 2 fair
**Presentation:** 2 fair
**Contribution:** 2 fair
**Rating:** 5
**Confidence:** 4

**Summary:**

This paper proposes to partition a sparse but connected (causal) graph into independent set and auxiliary set. Using this method of partition, treatment can be designed to estimate direct and spillover effects for causal inference tasks. Theoretical guarantees on bias/variance of the estimators were given together with simulation results.

**Strengths:**

1.	The problem definition is clear with good illustration to explain the concept of independent set and auxiliary set.
2.	Theoretical results are provided with good descriptions of the assumptions and limitations.

**Weaknesses:**

1.	The main weakness in this paper is the lack of a clear comparison to related works both theoretically and numerically. For example, how does the new theoretical guarantees improve over previous works? Otherwise, the analysis looks like an application of linear regression estimator.
2.	One contribution the paper claimed is using fewer assumptions for this model, it would be better to describe this more clearly. For example, what assumptions can be removed compared to previous works?
3.	The results rely on the greedy algorithm 1 to have a decent performance. Theorem 1 only gives the lower bound on ER graph which seems to limit the application of this framework.

**Questions:**

1.	The simulation results are comparing only to completely randomized design. Is it possible to compare with other designs cited in the introduction and Table 1?
2.	There are some typos, for example, in section 4.3 and 4.3.2, it is referring to section 2.3 and 2.3.2 which does not exist. (Should be 3.3 and 3.3.2?)

---

> ### Author Response · Authors · 2023-11-23
>
> We appreciate the reviewer for his/her detailed and insightful comments. We list our point-to-point responses to the weaknesses and questions:
>  1.  We listed a few other designs in the manuscripts for comparison, which includes the completely randomized design (CRD), randomized saturation design/graph clustering design (RSD), ego-clusters design (ECD). Neither of CRD or RSD is suitable for the networks described in the manuscript, which are well, though sparsely, connected. The reasons are (1) in CRD, most of the vertices receive an interference level as 1/2, resulting in a huge variance in the regression estimator;  (2) RSD requires isolated clusters — otherwise, significant bias arises. Only the ECD aims to solve a similar problem. However, as we illustrated in Table 1, at least for Erdos-Renyi random graphs, ECD is expected to have a smaller sample size compared to our methods. In summary, compared to CRD and RSD, which were developed for other problems, our method guarantees a smaller bias and smaller variance. Compared to ECD, our method achieves a larger effective sample size, resulting in a smaller variance (see e.g. Thm 3).
> 2. As we mentioned in the above response, compared to the well-known CRD, we do not require SUTVA, that is, we allow for interference. Compared to RSD (which is widely used in networks with interference), we relax the partial interference assumption to a heterogenous version in Assumption 1, and we do not require an isolated clustering structure of the network for the estimator to be unbiased.
> 3. We admit that the current lower bound is developed for Erdos-Renyi random graph. The distribution of the sampled network is determined by both the populational network and the sampling procedure, which is way too complicated to be discussed in this manuscript. Therefore, we establish the current partition result (Thm 1) on Erdos-Renyi network, while other theorems remain valid for other networks. Performance on other common random networks are evaluated through simulation instead.
> 4. Ego-clusters didn’t optimize the selection of egos and can be only applied to estimating spillover effect. To further illustrate the performance of Ego-clusters, we have included new simulation results of Ego-clusters design to the revised manuscript pdf in the updated Table 2 and Figure 2.
> 5. Thanks for pointing out the inconsistency of the section numbers, we modified the section numbers.

---

### Official Review · Reviewer_pw71 · 2023-11-01

**Soundness:** 3 good
**Presentation:** 3 good
**Contribution:** 3 good
**Rating:** 6
**Confidence:** 2

**Summary:**

The paper tries to find a lower bound on the random algorithm to find independent sets in Erdos-Reyni random graph. The paper claims that this independent set is of the order of the size of all nodes in the random graph. They further go on to use this result to estimate the bias and variance for direct effects and spillover effects related to their specific problem setup. They seem to further verify these results through computer simulations of an Erdos-Reyni graph. The results seem interesting to me. Specifically, I like Theorem 1 and based on my prior experience with Erdos-Reyni graphs, the results of this theorem seem to be intuitively correct. However, unfortunately I did not put the effort to follow the proofs in the paper in detail so I cannot independently verify their claims.

**Strengths:**

If the claim in Theorem 1 is correct that is an interesting result. Intuitively that result makes sense to me. However, I did not completely verify the proofs.

The paper presentation is good and readable.

**Weaknesses:**

I am not sure about the validity of assumptions used in the paper. Specifically assumptions 1 and 2. I would like to see more reasoning from the side of the authors on why these assumptions make sense. Any motivating examples could help the reader on these assumptions.

In equation 1, why is interference from neighbors simply summed up without any gains? Could it be the case that the interference from different neighbors can have a different effect on the results and we need to put more emphasis on some interference while putting less emphasis on other types of interference?

I suggest the authors to emphasis more in the paper that these results are derived for an Erdos-Reyni random graph setup and not necessarily any network. For instance, I did not see any mention of that in their abstract. The wording throughout the paper needs to be changed to reflect that these results are derived for random graphs.

 I would like the authors to specify in more detail that which part of their results is coming from different sources. For instance, Can the authors mention their contribution over Karwa & Airoldi (2018) in more detail?

**Questions:**

I am not sure about the validity of assumptions used in the paper. Specifically assumptions 1 and 2. I would like to see more reasoning from the side of the authors on why these assumptions make sense. Any motivating examples could help the reader on these assumptions.

In equation 1, why is interference from neighbors simply summed up without any gains? Could it be the case that the interference from different neighbors can have a different effect on the results and we need to put more emphasis on some interference while putting less emphasis on other types of interference?

I suggest the authors to emphasis more in the paper that these results are derived for an Erdos-Reyni random graph setup and not necessarily any network. For instance, I did not see any mention of that in their abstract. The wording throughout the paper needs to be changed to reflect that these results are derived for random graphs.

 I would like the authors to specify in more detail that which part of their results is coming from different sources. For instance, Can the authors mention their contribution over Karwa & Airoldi (2018) in more detail?

---

> ### Author Response · Authors · 2023-11-23
>
> We thank the reviewer for the detailed comments and would like to clarify the weakness the reviewer commented in a point-to-point fashion.
>
> 1.
> - Assumption 1 (interference on the proportion) is a very common assumption in experimental designs over networks (see Ugander et al., 2013; Forastiere et at., 2021; Cai et al., 2022). The assumption has two advantages. (1) it simplifies the heterogeneous interference effects from neighbors to a summary statistics (the proportion of treated neighbors) (2) it ensures that the interference received from neighbors are bounded as the proportion is a number between 0 and 1.
> - Assumption 2 (super-population perspective) is common as well (see, for example, Section 7.3 in Imbens and Rubin (2015)). Such an assumption is practical because in real experiments, a small sample of units as well as the underlying network of interference is drawn from the population in order to generate efficient estimations under a limited budget. On the other hand, since there always exist certain exotic networks where the proposed method does not work, our estimator aims to estimate the population causal effect instead of the sample one (with the given network). A similar setup for super-population perspective can be found in Shuangning and Wager (2022). Our setup suffices for the purpose of this paper.
>
> 2. We thank the reviewer for pointing out this issue, there may exist heterogeneous interference. To reduce model complexity, we assume interference only depends on the proportion of treated neighbors, we will leave heterogeneous interference in future work.
>
> 3. Thanks for the suggestion, we updated our abstract and other sections to reflect the results are derived for ER random graph.
>
> 4. Karwa & Airoldi (2018) can be considered as a special case of our approach. The design of Karwa & Airoldi (2018) could be only used to estimate the direct treatment effects and has no optimization. Our method could be implemented to estimate both the direct treatment effects and the spillover effects, and we optimize the result to find the largest independent set and reduce the variance of the estimation.
>
> Reference:
>
> - Johan Ugander, Brian Karrer, Lars Backstrom, and Jon Kleinberg. Graph cluster randomization: Network exposure to multiple universes. In Proceedings of the 19th ACM SIGKDD International Conference on Knowledge Discovery and Data Mining, KDD ’13, pp. 329–337, 2013.
> - Vishesh Karwa and Edoardo M Airoldi. A systematic investigation of classical causal inference strategies under mis-specification due to network interference. arXiv preprint arXiv:1810.08259, 2018.
> - Laura Forastiere, Edoardo M Airoldi, and Fabrizia Mealli. Identification and estimation of treatment and interference effects in observational studies on networks. Journal of the American Statistical Association, 116(534):901–918, 2021.
> - Chencheng Cai, Jean Pouget-Abadie, and Edoardo M Airoldi. Optimizing randomized and deterministic saturation designs under interference. arXiv preprint arXiv:2203.09682, 2022.
> - Yuchen Hu, Shuangning Li, Stefan Wager, Average direct and indirect causal effects under interference, Biometrika, Volume 109, Issue 4, December 2022, Pages 1165–1172.
> - Imbens, G. W. and Rubin, D. B., Causal Inference for Statistics, Social, and
> Biomedical Sciences: An Introduction, Cambridge University Press, 2015.

---

### Meta-Review · Area_Chair_KLfV · 2023-11-22

**Metareview:**

The paper addresses interference in causal experiments over social networks by proposing a novel experimental design based on independent sets. This design focuses on controlling interference exposures through treatment assignments, enhancing causal estimator performance by prioritizing sample quality over quantity. The approach demonstrates efficacy across various causal inference tasks, surpassing conventional methods, and is supported by empirical performance in simulations.

The key strength include: The paper provides a novel algorithm for treating units to minimize the effects of interference and maximizing sample efficiency for specific graph classes. Theoretical results are provided with good descriptions of the assumptions and limitations.

The key weakness:
There are some concerns over validity of assumptions, in particular, Assumption 2.

**Justification For Why Not Higher Score:**

The reviewers express a somewhat tepid positive reception to the paper.

**Justification For Why Not Lower Score:**

The paper has no significant concerns following the authors' rebuttal.

---

### Decision · Program_Chairs · 2024-01-16

Accept (poster)